# Implementation of Analog Perceptron as an Essential Element of Configurable Neural Networks

**DOI:** 10.3390/s20154222

**Published:** 2020-07-29

**Authors:** Chao Geng, Qingji Sun, Shigetoshi Nakatake

**Affiliations:** Department of Information and Media Engineering, The University of Kitakyushu, Kitakyushu 808-0135, Japan; y7dcb401@eng.kitakyu-u.ac.jp (C.G.); z8mcb401@eng.kitakyu-u.ac.jp (Q.S.)

**Keywords:** neural network, chip testing, analog perceptron, multi-layer perceptron, ReLU activation

## Abstract

Perceptron is an essential element in neural network (NN)-based machine learning, however, the effectiveness of various implementations by circuits is rarely demonstrated from chip testing. This paper presents the measured silicon results for the analog perceptron circuits fabricated in a 0.6 μm/±2.5 V complementary metal oxide semiconductor (CMOS) process, which are comprised of digital-to-analog converter (DAC)-based multipliers and phase shifters. The results from the measurement convinces us that our implementation attains the correct function and good performance. Furthermore, we propose the multi-layer perceptron (MLP) by utilizing analog perceptron where the structure and neurons as well as weights can be flexibly configured. The example given is to design a 2-3-4 MLP circuit with rectified linear unit (ReLU) activation, which consists of 2 input neurons, 3 hidden neurons, and 4 output neurons. Its experimental case shows that the simulated performance achieves a power dissipation of 200 mW, a range of working frequency from 0 to 1 MHz, and an error ratio within 12.7%. Finally, to demonstrate the feasibility and effectiveness of our analog perceptron for configuring a MLP, seven more analog-based MLPs designed with the same approach are used to analyze the simulation results with respect to various specifications, in which two cases are used to compare to their digital counterparts with the same structures.

## 1. Introduction

Artificial neural network (ANN)-based machine learning, known as a promising technology, has been researched widely to enable electronic devices more intelligent and efficient [1,2,3,4]. ANN is inspired by the brain of living creatures, which contains components such as neurons, connections, weights, and a propagation function. ANN has a huge set of neurons that are highly interconnected and arranged in layers. The structure of an artificial neuron mimics the function of dendrites, soma, and axon. Perceptron is an essential element of ANN, which has multiple inputs and a single output. It is commonly represented by a mathematical model as follows: Each input is multiplied by a weight and all weighted inputs are summed at the output, the resulting sum is then passed through an activation function. ANN-based machine learning fits the transfer function of the system by a training process where input-output pairs are iteratively presented, while the variable parameters/weights are adjusted with the process.

Many digital- or analog-implemented circuits for perceptron have been proposed in the literature, showing good results in the simulation phase [5,6,7,8], however the drawback of these works is the lack of silicon measurement results, which are of a great importance for the investigation of the fundamental characteristics of a perceptron circuit. Besides, the multi-layer perceptron (MLP) is constituted by perceptron, which is a fundamental structure for the feedforward neural network (NN), in VLSI (very-large-scale integration) implementations incorporating various learning algorithms, thus making MLP a common choice as it has been continuously researched for many years [9,10,11,12,13,14]. In the state-of-the-art works, the authors in [15] implement a low-latency MLP processor for real-time cancer detection using field programmable gate arrays (FPGAs), under mass-spectrometry benchmarks, it outperforms both the central processing unit (CPU) and graphics processing unit (GPU) implementations, with an average speedup of 144× and 21×, respectively. The work [16] presents FPGA-based electrocardiogram (ECG) arrhythmia detection by using principal component analysis for feature reduction and a MLP for classification with an accuracy of 99.82% achieved on average for the MIT-BIH database. A recent work implemented on FPGA can also be seen in [17] where an efficient MLP designed with 16-bit fixed-point data precision is used for human activity recognition and to obtain a high recognition accuracy.

On the other hand, the analog MLP has been popular in recent years for NN-based machine learning. Several analog MLPs have been implemented to solve classification problems in the past decade, which show promising results. One can see a recent work in [18], where the metabolic adders and perceptrons presented are the first engineered biological circuits that use metabolism for analog computation, the authors highlight the higher efficiency of analog computation on a continuous input compared with digital logical circuitry. In [19], the authors demonstrate a one-hidden layer perceptron classifier circuit based on passive memristive crossbars for pattern classification, where they highlight the higher energy efficiency and neuromorphic performance of the anaolg and mixed-signal integrated circuits, compared with their digital counterparts. In [20], an analog MLP with programmable neuron is introduced that achieves the accuracy of more than 99.9% for XOR gate implementation and a 48% reduction in power consumption compared with previous works. The authors emphasize that digital NNs have high accuracy, low speed, high chip area, and high power consumption whereas analog NNs are optimized in chip area and processing speed as well as power consumption. A precise quantitative comparison in neuron implementation between digital and analog designs can be found in [21], in the same technology node and the same level of performance, the analog implementation requires five times less area and consumes 20 times less energy than the digital design.

Aiming at the advanced sensor nodes, such as the biological sensor systems introduced in [22,23], Ishiguchi (our group member) et al. has proposed an analog perceptron circuit with a DAC-based multiplier in [24]. In comparison with the traditional sensor system where the central processing unit and signal processing unit are necessary. Analog VLSI implementations are preferable for a sensor node as they have lower power and a smaller device area. The preliminary results provided in [24] is the base of developing our research. In this work, an analog perceptron comprising of DAC-based multipliers and phase shifters is fabricated in a 0.6 µm/±2.5 V CMOS process. The measurement results of the chip are reported as well, which shows good functionality and performance for circuits. In addition, the noise issue arising from the circuit and the accuracy issue of the resistor have been discussed.

Furthermore, after the fundamental characteristics of our analog perceptron is verified, we propose the MLP-based neural network by utilizing analog perceptron, where the structure and neurons as well as weights can be flexibly configured. Since the MLP realized by analog circuits is vulnerable to many factors, such as offset voltage, noise coupling, impedance, limited scalability, and process-introduced variation [20,25,26], these accumulated effects degrade the feasibility of the MLP circuit. Besides, most of the analog MLP implementations adopt complicated circuits for the activation function, which are difficult to train, and in addition the weights are not easy to control. After tackling these challenging problems, this paper presents a successful implementation of a 2-3-4 analog-based MLP circuit. We design the MLP circuit with rectified linear unit (ReLU) activation in a 0.6 µm CMOS technology process with supply voltages of ±2.5 V. The circuit utilizes an improved source follower with a simple structure to approximate the ReLU function greatly. Since an operational amplifier (OPAMP)-based adder and an inverting amplifier are inserted in each perceptron, the DC biasing of the ReLU circuit following the perceptron is adjustable, so that the reliability of the whole circuit is improved. In addition, we present the impedance issue in the cascading of neurons in the MLP, which is critical to the feasibility of the whole circuit. Finally, seven more analog-based MLPs are designed using the same way as in a 2-3-4 MLP, where the structure and neurons are selected under the geometric rule and rounding rule. We summarize simulation results with respect to various specifications and make a comparison to FPGA-based MLPs with the same structures, which show the feasibility and effectiveness to construct a MLP-based neural network by utilizing our analog perceptron.

The rest of the paper is organized as follows. Section 2 introduces our motivation for analog-based MLP implementation and configurable neural networks based on analog perceptron. Section 3 describes the perceptron chip and a measuring system for the circuits as well as the measured results. Section 4 presents configurable neural network by utilizing analog perceptron and introduces some critical aspects to the success of the implementation of the MLP circuit. Section 5 explains the simulation of the whole MLP circuit and summarizes the simulation results. Finally, Section 6 concludes this work.

## 2. Preliminary

In this section, we present an idea of Fourier series approximation in the time domain aiming at electroencephalogram (EEG) signal processing and also show the advantages of analog-based MLP as compared to FPGA-based MLP. Our motivation for analog-based MLP implementation is introduced. Subsequently, we introduce the process to implement the configurable neural networks based on analog perceptron, which consists of the choices for the analog-based MLP structure and an activation function as well as a learning algorithm. Furthermore, we highlight that the main contributions of this work are the measurement of the perceptron chip and the circuit simulation for MLP physical hardware.

### 2.1. Motivation for Analog-Based MLP Implementation

EEG signals can be measured from the surface of the head using appropriate electrodes and instrumentation. The signals are time-varying patterns of electrical energy that are typically in the microvolt range, which are reflections of underlying human brain states and thus be used for medical diagnosis. On the other hand, according to the theory of the Fourier series, a period (or close to period-fashion) signal is constituted by the sum of a series of weighted sine or cosine waves. An arbitrary signal in that interval (or the entire signal if it is period) can be approximately synthesized by a period of summation with appropriate weights.

For periodic signals on unbounded intervals, the Fourier series aims to use a series of sine waves or cosine waves of different magnitudes and frequencies, along with a method of choosing the appropriate weights, so as to produce an approximation to the original signal. The error between the true signal and its estimate can be reduced as more terms (sine and cosine waves) are added. The terms can be completely described if we know their magnitudes and frequencies as these are very convenient basis functions. The basis functions and weights that describe a given signal can be derived from the Fourier transform. Inspired by this, we attempt to use the Fourier series incorporating a machine learning mechanism to decompose or reconstruct the raw EEG signal in the time domain, as shown in Figure 1a. Note that pre-processing is necessary for the training dataset when employing our machine learning, we can truncate a certain period of EEG signal and repeat it with multiple times, so as to turn it into a signal close to periodic fashion.

In a conventional way to extract the feature of the EEG signal for evaluating the brain states, electrodes (analog front-end) for EEG signal acquisition and analog signal processor (analog back-end) for post-processing and digitization are typically employed [22], and then digitized signals are processed in the frequency domain by software, where the Fourier series approximation is implemented by a process referred to as the fast Fourier transform (FFT). Finally, the feature of an EEG signal such as, spectrum, frequency components, and power can be extracted. However, the whole system from EEG signal acquisition to feature extraction is very complicated and expensive. Therefore, our idea is to use the neural network to realize the Fourier series approximation in the time domain directly, as the neural network has multi-input or multi-output which is appropriate to the situation. There is an analog method and digital method for choosing, as shown in Figure 1b.

For the same target of signal decomposition, however, FPGA-based MLP needs more hardware than analog-based MLP since it can only deal with discrete-time signal, back-end processor with analog-to-digital converters (ADCs) and digital-to-analog converters (DACs) are needed in its preceding and succeeding stages, respectively. Compared with its counterpart, analog-based MLP, FPGA-based MLP is not only expensive in hardware cost but also has a lower efficiency in learning algorithms and consumes more power.

Suppose that there is a three-layer m-h-n (m are input neurons, h are hidden neurons, and n are output neurons) for the MLP whose neurons are fully connected, it needs to do (h×m+n×h) times of multiplication and (h+n) times of addition for the signal propagation from the inputs to outputs of the network. For analog-based MLP, it needs (h+n) multipliers and (h+n) adders, the total power consumption is that of multipliers and adders as well as activation circuits. Its outputs nearly respond to inputs synchronously. For FPGA-based MLP under the assumption of input analog signal frequency *f*, ADC with a resolution of *X* bits, clock frequency $f_{CLK}$, sampling frequency $f_{s}$, the bit number of connection weight, *Y* bits (Y≤X), and the propagation delay time from the inputs to outputs of MLP at each sampling period, $t_{d}$. The FPGA-based MLP needs (h×m+n×h) *X*-bit multipliers and (h+n) 2X-bit adders. The total power consumption is that of all logical modules. Note that the above assumption uses a simple-structure MLP with a single hidden layer for both methods, compared with analog-based MLP, the logical modules of FPGA-based MLP increase more drastically as the hidden layer number and neuron number increase.

Furthermore, for FPGA-based MLP, the total delay *t* at a period of the input analog signal can be formulated as follows: $t=X\times \left(f/f_{s}+1\right)\times t_{d}$. We take a specific case as an example, for an analog signal *f* = 1 Kz, an 8-bit ADC with $f_{CLK}$ = 10 MHz, $f_{s}$ = 1 MHz, a FPGA-based MLP with 8-bit connection weights, and $t_{d}$ = 100 ns. By applying the above formula of the total delay at a period, it yields *t* which is around 0.8 ms, meaning its output is 0.8 ms behind the analog-based MLP output at each signal period and its total delay over the whole training time becomes very large compared with the analog-based MLP. Note that the result is under the consumption of the best case and without considering DAC conversion. Thus, analog-based MLP shows a higher efficiency than FPGA-based MLP when deployed in machine learning.

### 2.2. Configurable Neural Networks Based on Analog Perceptron

The process to implement the configurable neural networks is shown in Figure 2a. The network employs the analog-based MLP structure where all neurons are fully connected. The structure connected in the red line is an example of 2-3-4 MLP. The structure connected in the black-dashed line represents other candidate structures within the available hardware resources, which shows the possibility of exploring different structures depending on the hardware resources and the needs. The neuron is constituted by one or multiple DAC-based multipliers, and more multipliers are needed when the input ports of a neuron are not sufficient.

The activation module can be implemented by simple analog circuits to achieve common functions such as Sigmoid, tanh, and ReLU. Among these functions, the standard ReLU shows more advantages than other functions in terms of the gradient vanishing problem, convergence speed, and computation efficiency. Besides, the standard ReLU is the most popular and promising activation function in convolutional neural networks (CNNs) and deep neural network (DNNs) as it shows the more generalization ability and ability to deepen the hidden layer, which are important for a complex EEG signal processing by using analog-based MLP. Last but not least, we can approximate the standard ReLU function by using an improved source follower with simple devices, which is more appropriate to our case as other activation circuits need design specifically. Even though compared with the standard ReLU, the employed ReLU circuit in this work decreases a bit of advantage due to the use of capacitor and resistor, it has achieved a great approximation targeting at the standard ReLU. However, our analog-based MLP maintains the possibility of exploring other activation functions as the network is configurable.

As for the learning algorithm, we can use a publicly-available EEG dataset as our training dataset, but we need to pre-process the EEG signals of the dataset to make them close to periodic and then the processed EEG signals are decomposed in the time domain by the Fourier series approximation, and the decomposed signal components with different frequencies and amplitudes are used to compare to the predicted results of our neural network. The whole process for dataset preparation can be realized by an efficient algorithm. Next, the MLP software is implemented in the TensorFlow which is an open-source machine learning platform. The choice for the structure and neurons in each layer of the MLP commonly uses the geometric pyramid rule as referred in [27].

For a three-layer network with *n* input neurons, *m* output neurons, the hidden layer has $\sqrt{n\cdot m}$ neurons. For the networks with *n* input neurons, *m* output neurons, and *i* (i≥2) hidden layers, a basic parameter *r* for the hidden layers (from $h_{1}$ to $h_{i}$) can be expressed as: $r={\left(n/m\right)}^{1/i+1}$. Thus, the neuron number of the *i*-th hidden layer can be expressed as: $h_{1}=m\cdot r^{i}$, $h_{2}=m\cdot r^{i-1}$, …, $h_{i}=m\cdot r$. The input neuron number *n* and the output neuron number *m* need to be set based on actual needs. The geometric pyramid rule also applies to this work but with a slight difference, when the neuron number derived from the rule is not an integer, we need to round the number. For instance, a 2-3-4 MLP demonstrated herein yields 2.83 for the hidden neuron number but it is rounded to 3. We set a constraint for the learning algorithm to limit the total number of neurons, the layout area for neurons cannot exceed the sum of all the layout space available, the upper bound is the die size that is assigned for neurons. Thus, the learning algorithm enumerates all the candidate structures under the constraint and can avoid the infinite loop. Besides, the backpropagation (BP) algorithm is based on the Levenberg–-Marquardt (LM) algorithm [28], which computes the error between the true values (frequencies and amplitudes) and the predicted values, and then updates the gradient and weights to minimize the error. Note that our analog-based MLP hardware uses off-line learning manner, i.e., we first implement an optimal MLP model in the software and determine its structure and weights, and then implement analog-based MLP in hardware whose structure and weights are consistent with software.

Once a neural network model is trained by the back-propagation algorithm in the software, the weight value of each connection can be determined. As long as it is ensuring the configuration and weights are consistent with the software, our circuit can be used to recover a learning model in hardware. In this paper, we primarily focus on the implementation of the circuit and verify the feasibility of configuring analog-based MLP by perceptron, the software part is beyond the scope of this paper.

By utilizing the analog perceptron circuits and activation circuits with some electronic components, after fine-tuning, we can construct analog-based MLP hardware for a learning model obtained from the software. The reasonability of the number of layers and neurons per layer as well as component values can be verified by circuit simulation. If the simulated result for the circuit is unsatisfactory, we can iterate the process until determining a network and weights suitable for our requirements. Once a learning model is obtained and each weight of connection is determined, as well as a fine-tuned analog-based MLP hardware is implemented, the digital circuit sets each weight of the MLP circuit to a corresponding value consistent with the software. We configure the neurons of our analog-based MLP by digital codes of DAC-based multipliers, so that the effectiveness of the learning model can be verified in hardware. In this manner, the configurability and flexibility of our method are improved, which is helpful for our circuit devoted to analog front-end of the biomedical sensor.

Our contributions in this work are illustrated in Figure 2b. The feasibility of MLP hardware implementation is most essential to our final aim, thus this work mainly focuses on the measurement of the perceptron chip and the circuit simulation for MLP physical hardware, so as to find the issue of interconnecting perceptron circuits and optimal parameters for activation circuits. Physical hardware implementation of analog-based MLP for a given learning model is a future work, since the learning algorithm in the software still needs to be optimized.

## 3. A Measurement-Verified Perceptron Chip

This section introduces a measuring system which is comprised of software and hardware. The schematic and layout for the circuits under test are used to detail the implementation of our analog perceptron. Finally, we report the measurement results by which the fundamental characteristics of analog perceptron are verified and the effectiveness of the perceptron chip is demonstrated. We analyze the noise issue and the resistance accuracy issue, and propose solutions to tackle these issues.

### 3.1. A Measuring System for Perceptron Chip

Aimed at an analog front-end of a sensor node, Ishiguchi et al. propose an analog perceptron circuit with DAC-based multiplier in the work [24] to improve the sensing information. An analog perceptron circuit with a DAC-based multiplier (namely, Cir.01.) and DAC-based perceptron circuit with phase shifters (namely, Cir.02.), have been fabricated to validate the effectiveness of circuits. The general architecture of the measuring system for our perceptron chip is illustrated in Figure 3. The chip is fabricated in a 0.6 µm/±2.5 V CMOS process and mounted on a custom printed circuit board (PCB). All the chip pins are carefully connected by wires to the ports of the experimental setup, analog discovery 2, which is a portable and powerful device with high performance comparable to a stack of traditional measurement instruments, such as oscilloscope, arbitrary waveform generator, power supply, digital pattern generator, etc. With a software running on a computer, WaveForms, the device can be configured to measure, visualize, record, and control mix-signal circuits of all kinds. In our measurement, oscilloscope and arbitrary waveform generator are capable of providing sample rate of up to 100 MS/s and a bandwidth of up to 12 MHz. Two power supplies ranging from −5 V to +5 V can satisfy the measurement requirement for the power source. A digital pattern generator can also provide sufficient channels to generate or visualize patterns for the test chip.

### 3.2. Schematic and Layout for the Circuits under Test

The analog perceptron circuit with DAC-based multiplier employs the mechanism of neural network-based machine learning, the model of which can be represented as: $f_{out}\left(t\right)=w_{1}\cdot f_{1}\left(t\right)+w_{2}\cdot f_{2}\left(t\right)+\cdots +w_{n}\cdot f_{n}\left(t\right)$. Where $f_{1}\left(t\right),\cdots ,f_{n}\left(t\right)$ are the inputs, each of which is multiplied by a weight $w_{1},\cdots ,w_{n}$, all weighted inputs are summed at the output $f_{out}\left(t\right)$. All the weights are controlled only by digital codes. Two DACs are used in a DAC-based multiplier, a DAC is inserted at the input of the negative feedback of the OPAMP circuit, while the other is at the loop of the feedback. DACs serve as two variable resistors and a changing of the DAC output current looks as if the resistance value were to change. The input-output/$V_{in}-V_{out}$ relationship of the multiplier can be represented as: $V_{out}=-\frac{X1}{X4}V_{in}$, where X1 and X4 are the decimal input codes of DAC1 and DAC4, respectively. For a three-input perceptron circuit, the input-output/$V_{in}-V_{out}$ relationship of the circuit can be represented as: $V_{out}=-\frac{X1}{X4}V_{in1}-\frac{X2}{X4}V_{in2}-\frac{X3}{X4}V_{in3}$, where X1, X2, and X3 are the decimal codes of DACs for the input $V_{in1}$, $V_{in2}$, and $V_{in3}$, respectively. While X4 is the decimal code of DAC in the negative feedback of the OPAMP circuit.

In addition, Ishiguchi et al. also introduces phase shift circuits to the perceptron model to deal with time series inputs. It is applicable to prediction algorithms such as the recurrent neural network and chaos pass filter. The only difference from Cir.01 is that one phase shifter is inserted between inputs, $V_{in1}$ and $V_{in2}$, while the other phase shifter is inserted between inputs, $V_{in2}$ and $V_{in3}$. A model of time series by Cir.02 can be formulated as follows: $f_{out}\left(t\right)=w_{1}\cdot f\left(t\right)+w_{2}\cdot f\left(t+d\right)+w_{3}\cdot f\left(t+2d\right)$, where *t* and *d* are time and delay variables, respectively. In their work, by applying different component values for different frequencies in the phase shifter, Cir.02 can achieve that the phase delay is evenly generated by 90° among the multiplier output signals. That is because phase shifters can control the absolute delay.

Figure 4 shows a schematic of a perceptron chip and block diagram of the measurement. Our perceptron chip consists of two circuits, Cir.01 and Cir.02. When measuring the former, phase shifters are disconnected and excluded from the Cir.01. When measuring the later, phase shifters are connected and included in the Cir.01 by connecting the peripheral pins of the chip. In the figure, phase shifters enclosed by the dashed box and the inputs of Cir.01 are in virtual connection, it means that phase shifters are disconnected to Cir.01 when Cir.01 is in measurement and phase shifters are connected to Cir.01 when Cir.02 is in measurement. Thus, the layout area can be smaller, and also the number of pads to be occupied can be reduced.

In the circuit under test, there is a sum module following after the outputs of three multipliers, which is constituted by an OPAMP-based adder and an inverting amplifier. The whole circuit is slightly different from that reported in the work [24]. This is because in the previous design, it is found that the feedback current noise arising from the outputs of DACs can affect the biasing of each other, leading to the waveform distortion in the final output. Therefore, we assign three multipliers separately and collect their outputs by a summator, rather than combining the outputs of DACs at the inverting input of the OPAMP. Another benefit of the improved design is that we can obtain the inverting and non-inverting output simultaneously, which is useful for negative and positive weight generation. There are two types of phase shifters in the circuits for input frequencies 1 KHz and 10 KHz, respectively. Each type has two identical phase shifters as shown in the schematic. Phase shifter requires different component values for different frequencies. At 1 KHz, the resistance values for $R_{4}$, $R_{5}$, and $R_{6}$ are all 1.55 MΩ, the capacitance value for $C_{1}$ is 100 pF. While at 10 KHz, the resistance values are all 155 KΩ and the capacitance value is 100 pF.

In the periphery of the chip, our experimental setup, analog discovery 2, provides testing signals for the inputs of the perceptron circuits. The digital pattern is provided for the digital codes of DACs to control the weight of each analog input. The oscilloscope is used to measure the analog signal nodes of interest. I/O pads serve as the interface between the experimental setup and chip. Finally, the power supply provides supply voltages of ±2.5 V for the I/O pads and the whole circuit. Two decoupling capacitors in parallel with power supply are also used to reduce the power noise.

The top-level layout of the perceptron chip is shown in Figure 5. The layout is implemented in a 0.6 µm CMOS process with supply voltages of ±2.5 V. The total area is 1.6 × 1.6 ${\mathrm{mm}}^{2}$ and the active area excluding pads and empty space is 1.01 ${\mathrm{mm}}^{2}$. Note that the two phase shifters for the same frequency are combined into one in the layout. One can see the layout, for the input frequency 1 KHz, two identical phase shifters are implemented that are symmetrical with respect to the horizontal line; for the input frequency 10 KHz, two other phase shifters are implemented that are symmetrical with respect to the vertical line. Phase shifters for Cir.02 occupy the most area due to the large capacitors and resistors, whereas the area of the Cir.01 is only around 0.17 ${\mathrm{mm}}^{2}$. Cir.01 is comprised of three DAC-based multipliers and an analog summator whose topology is fully consistent with the schematic in Figure 4. Cir.02 is comprised of Cir.01 and phase shifters, the connection of Cir.01 and phase shifters depends on the input frequency. When Cir.01 is in measurement, phase shifters are disconnected to Cir.01. When Cir.02 is in measurement, phase shifters are connected to Cir.01 by the peripheral pins of the perceptron chip. Furthermore, the fabrication process is a three-layer metal process with a single-well. The p-channel device is formed within n-well in the p-type substrate while the n-channel device is formed directly in the substrate.

### 3.3. The Measurement Result and Analysis

The resolution of the measurement instrument is a critical consideration for a high-accurate measurement for the circuit, our experimental setup can provide a high-resolution thanks to its resolution of up to 100 uV per division. Noise reduction is another critical consideration in the field of measurement [29,30,31,32,33]. A clean target signal without significant noise interference can improve the measurement, thereby obtaining a reliable evaluation on the circuit. However, it is a challenging problem for the analog circuit measurement in the presence of noise, especially for the low-frequency and small-signal circuit, which can be easily overwhelmed by noise. The circuit noise generally contains the external part induced by electromagnetic interference and power supply, and also contains the intrinsic part such as thermal noise, flicker noise (i.e., 1/f noise.), and shot noise. In our measurement for the experimental case, the method for the electromagnetic shielding was employed and decoupling capacitors were connected in parallel with the power supply to reduce voltage fluctuation. After dealing with the external noise issue, large numbers of experiments were performed among a set of chips. All the experimental chips obtained identical measured results, which are shown in Figure 6.

In the measurement for Cir.01 shown in Figure 6(a-1), there are three input sinusoidal signals Vin1, Vin2, and Vin3, with an amplitude of 5 mV, 10 mV, and 15 mV at 1 KHz frequency, respectively. The multiplication factors (i.e., weights.) of three multipliers are set to be 3, 2, and 1, respectively. Therefore, the weighted-sum amplitude of the final output at the Vout-node should be 50 mV. However, the intrinsic noise from the circuit affected the final output so that the amplitude is actually 57.2 mV. After inserting a two-order Butterworth low-pass filter with the cut-off frequency of 1 MHz at the Vout- node, the effect of noise coupling is significantly alleviated, and the amplitude is 49.7 mV as compared to the desired value 50 mV. The frequency of all of the sin waves in the graph is 1 KHz, and the phase of the final output is inverting to that of the input. The offset of the final output is negligible and has only hundreds of microvolts, and all the sin waves are adjusted to the appropriate position so that they can be observed clearly. The measured results are consistent with the simulation results reported in the work by [24].

Furthermore, for comparison to the experimental case in Figure 6(a-1), various experiments with different settings are conducted to validate the circuit. Figure 6(a-2,a-3) shows the measurement results with the same amplitude and frequency but with different weights and Figure 6(a-4) shows the measurement result with the same frequency and weights but with different amplitudes. For a comparison to Figure 6(a-4), Figure 6(a-5) shows the measurement result with the same amplitude and weights but with different frequencies, and the shape of the waveform for the final output can change with the input frequency. Note that the measuring method for all of the conducted experiments is kept consistent as in Figure 6(a-1), and all the settings for noise filtering are the same. Therefore, all measured results above demonstrate that the Cir.01 could be correctly controlled by digital codes, the amplitude, and the frequency of the input signals, respectively. In terms of the amplitude of the final output, it can be verified by the weighted-sum calculation. After filtering the noise, the Cir.01 shows a good performance with respect to the consistency between the measured values and theoretical values.

The Cir.01 demonstrates the effectiveness of our design and it is applicable to the MLP model, though, we still measure the noise for the critical module. From the function of the noise spectral density of OPAMP, it is found that the total input-referred noise of OPAMP over the frequency from 0 to 20 MHz is 1.9uV/$\sqrt{\mathrm{Hz}}$. As compared to the low-noise OPAMP with tens of nV/$\sqrt{\mathrm{Hz}}$ within the same bandwidth, the specification of our OPAMP is disadvantageous. Since the noise of OPAMP is dominated by the white noise (i.e., thermal noise) and flicker noise (i.e., 1/f noise), thus the noise power can be reduced by a low-pass filter. In addition, since the chip is implemented in a single-well process, the substrate noise also contributes a portion of noise at the final output [34]. It suggests that in future works, the twin-well process and low-noise design technique such as chopper OPAM [35,36,37], can be used to further optimize the Cir.01.

As shown in Figure 6b, in the measurement result for Cir.02, we can observe that the phase delay is evenly generated by 90° among the output signals of multipliers. However, the measured result is inconsistent with the simulated result reported in the work [24]. All the weights of multipliers are set to be 1 for both the simulation and measurement. The input signal of 0.01-V amplitude and 10-kHz frequency is applied in the simulation, whereas the input signal in the measurement actually needs 0.1-V amplitude and 58-kHz frequency for the 90-degree phase delay. However, the post-layout simulation result is verified to be correct and has no discrepancy with the pre-simulation. According to the experience of circuit designers using the same manufacturing process, such a difference is mainly caused by the poor accuracy of the high-poly resistor in the process. Given that the phase delay of our phase shifter is formulated as: $d=-2tan^{-1}\left(2\pi fCR\right)$, where d is an absolute phase delay between the input and output signals, f is the input frequency, C is the capacitance value, and R is the resistance value. In a fabricated chip, both R and C are constant. Therefore, the function d monotonously decreases with the frequency, f. When setting the frequency f to be 10 KHz in our measurement, the measured phase delay is bigger than that of the simulated. Therefore, the actual resistance value is smaller than that of the simulated. When increasing the input frequency, the phase difference among the multiplier output signals is finally adjusted to 90°. However, due to the very large difference between the actual resistance value and theoretical resistance value for 1 KHz, the measurement for Cir.02 at 1 KHz failed to perform. Thus, only the measurement result for Cir.02 at 10 KHz is reported in this work. Nevertheless, our perceptron chip is applicable to prediction algorithms for the application targeting at 58 KHz signal.

## 4. An MLP Circuit for Configurable Neural Network

After fundamental characteristics of our analog perceptron are verified by measurement, as an extended work of perceptron, in this section, a multi-layer perceptron circuit with a non-linear activation function is introduced. We present a process to configure a MLP by utilizing analog perceptron, and also verify the effectiveness of the selected structure by circuit simulation.

A block diagram of the MLP circuit is shown in Figure 7. This network is constituted by three layers: An input layer with two neurons, a hidden layer with three neurons, and an output layer with four neurons. The neuron number is annotated in the corresponding module, the net name is annotated as well and the connection relationship is indicated by its name directly. Each neuron in the hidden and output layers is followed by a ReLU module, which is used to approximate the ReLU function in the neural network. The function of a perceptron and ReLU circuit can be represented by a mathematical model, as shown in the dotted box of the figure. Where $w_{ij}$ denotes the weight for the connection of two neurons. The subscript *i* represents the sequence number of the starting neuron, and *j* represents that of the target neuron. For example, the weights of the sixth neuron are $w_{36}$, $w_{46}$ and $w_{56}$, respectively. $x_{i}$ denotes the input of the *j*-th neuron, and $y_{j}$ denotes the output of the *j*-th neuron. The perceptron module is comprised of three or two DAC-based multipliers and other function circuits. The ReLU module is comprised of an improved source follower and an OPAMP-based buffer. The weight of each perceptron module is controlled by digital codes.

### 4.1. An Improved Source Follower

ReLU is the most commonly used activation function in neural networks, especially in CNNs and DNNs. Mathematically, it is defined as: y=max(0,x),x∈R. Inspired by the input-output characteristic curve of a source follower, we attempt to implement a ReLU circuit by since the curve is approximately similar to the ReLU function. However, the output voltage of a typical source follower (see Figure 8a) always has a significant difference with the input voltage, which reduces the accuracy of the neural network model and prediction correctness. Therefore, we modify the circuit and its configuration is shown in Figure 8b. The value of each component is annotated as well. When a proper DC operating voltage is applied at the input, the NMOS transistor works in a saturation condition. The DC operating voltage applied influences not only the drain current of M2, but also the linearity of input-output characteristic. Hence, a proper DC biasing is important to approximate the ReLU function. In the preceding stage of the ReLU circuit, we have an OPAMP-based adder that incorporates a DC biasing, which can adjust the DC operating point of the ReLU circuit. The input-output characteristic of the above two circuits is shown in the Figure 8c. The output of both circuits is at the source node of transistor. An improved source follower (b) can better approximate the ReLU function, as the difference between the input and output is smaller.

Note that both the gain of circuits in Figure 8 is not larger than 1, even if the R1 and R3 are close to infinite. When R3 reaches a certain value, increasing the value of R3 only contributes a little to the slope of the curve. Capacitor C1 is floating and prepared for AC coupling so as not to influence the DC operating point of the next stage. In order to increase the accuracy of the source follower, R2 is used to reduce the second-order effect which is channel length modulation. However, the voltage loss on the ReLU circuit is inevitable as the output cannot perfectly follow the input. We also note that when the input voltage for (b) is beyond 4.7 V, the output voltage does not increase and that is because as the gate-source voltage increases, the number of carriers in the transistor channel no longer increases, and the drain current tends to be constant. Last, we have to mention that transconductance amplifiers (OTAs) are good choices for resistors to reduce the silicon area and power consumption [38].

### 4.2. Summator for Improving Reliability

Inside the perceptron module of the top-level schematic, there are DAC-based multipliers and summators. The circuit shown in Figure 9 is a sub-circuit of a twp-input perceptron, which is constituted by an OPAMP-adder and an inverting amplifier. The port name and value of all resistors are annotated in the figure. In our implementation of the MLP circuit, V0 connects to ground, VIN1, and VIN2 connect to the outputs of multipliers, respectively. VBIAS connects to a biasing generator circuit that provides various biases needed. It can serve as a bias in the mathematical model of the neural network. By utilizing the bias, the centering voltage of the summed signals can be adjusted. Besides, the DC biasing can also be used to bias the ReLU circuit, which follows after the perceptron, so as to adjust its operating point. Note that the bias of each ReLU can be different and independent, therefore the reliability and flexibility of MLP is improved.

The inverting amplifier is used to invert the phase of the summed signals. As a perceptron circuit with a DAC-based multiplier always generates an inverting signal, such as Vout=(−X1/X2)Vin1+(−X3/X4)Vin2, here, *X* represents the decimal value of the digital codes of DAC. It is hard to verify the output of each neuron in this form, especially in a complicated neural network with multiple hidden layers. Therefore, we use an inverting amplifier to achieve that the output of neurons always has the same phase with the input, so that the calculation at each neuron becomes easy. Note that the positive port of the inverting amplifier connects to VBIAS rather than the ground. That is because the OPAMP-based adder has a DC component at its output. For a three-input perceptron, when necessary, we reserve a periphery pin for the non-inverting input of the OPAMP-based adder, so that the DC biasing can be coupled to that input through a resistor.

### 4.3. Impedance Issue of Cascading Neurons

Since the output impedance of the ReLU circuit is extremely high, therefore it has a poor ability to drive its succeeding stage. The high output impedance of the ReLU circuit results in not working when combining two neurons. Consequently, it is critical to deal with the impedance issue when cascading neurons. In our implementation of a MLP circuit, we measure the output impedance of a ReLU circuit following a two-input perceptron, and the input impedance of a three-input perceptron, which follows the ReLU circuit. As shown in the graphs of Figure 10a, Zout1 represents the output impedance of a ReLU circuit: Zin1, Zin2, and Zin3 represent the input impedance of three inputs, respectively and Zout2 represents the output impedance of the ReLU circuit inserting an OPAMP-based buffer after it. When the circuit works at 1 KHz, before inserting a buffer, the output impedance is very high, reaching 15.9 MΩ, whereas the input impedance of three inputs is 34.1 KΩ, 22.9 KΩ, and 18.3 KΩ, respectively. It is far smaller than the output impedance of the ReLU circuit, therefore ReLU cannot drive the next stage without dealing with impedance issue. In dealing with the impedance issue of cascading circuits, inserting a buffer between two stages is an effective way. As seen in the graph for Zout2, in our implementation of the circuit, the output impedance is significantly reduced after inserting a buffer, which has only 6.1 mΩ and load-carrying capacity has been improved.

However, it is not any buffer that can alleviate the issue of high output impedance. We need to design the output impedance of the buffer carefully. Figure 10b shows the measuring results of the output impedance of four buffers. There are OP1-based, OP2-based, OP3-based, and INV-based buffers. When working at 1 KHz, their output impedance is 335.1 Ω, 2.7 KΩ, 333.9 mΩ, and 1.8 KΩ, respectively. Finally, we chose the OP3-based buffer as it has the lowest output impedance. Besides, it has a stable frequency characteristic ranging from 0–1 MHz. However, the other three buffers cannot enable the MLP circuit to work, even inserting them after a ReLU circuit. Note that the ReLU circuit inserting with an OP3-based buffer is packed inside the ReLU module in the top-level schematic.

Furthermore, from both figures of the measuring results, it shows that the impedance of the circuit is relevant to the working frequency, as the impedance value of the circuit is changed dramatically when the frequency is beyond 1 MHz. It implies that our MLP circuit is constrained by the working frequency.

## 5. Experimental Case for MLP Circuit

After the 2-3-4 MLP is successfully constructed, in this section, we first explain the simulation settings for a 2-3-4 MLP, and then summarize the simulation results of a 2-3-4 MLP. Finally, seven more analog-based MLPs are also designed through the same method exampled in the 2-3-4 MLP, in which two cases are used to compared to their digital counterparts with the same structures. The simulation results relative to various specifications are summarized and analyzed, demonstrating the feasibility and effectiveness of our analog perceptron for configuring a MLP.

### 5.1. Simulation Explanation

After each module of the 2-3-4 MLP circuit is well tuned, we conduct a system simulation for the top-level schematic (see Figure 7). Since the experiment demonstrated in this work is used to verify the feasibility of configuring an analog-based MLP, and determine the most suitable circuit parameters, the connection weights in this experiment are preset by the digital codes of DAC-based multipliers. Subsequently, two test signals are applied to the inputs. Each measuring net of the MLP circuit is plotted and the amplitude of the waveform is recorded. Since the result of each measuring net is a simple weighted sum, the measuring result of simulation can be used to verify the behavior of a learning model on hardware easily, the effectiveness and feasibility of our MLP circuit can be demonstrated as well.

In our system simulation, we apply two of the same signals with ab amplitude of 1 mV and frequency of 1 MHz at inputs, and measure the points of interest on the nets of the top-level schematic. As seen in the Figure 11, we obtain five sets of graphs, corresponding to the measuring nets of inputs (see Figure 11a), neuron outputs of the hidden layer (see Figure 11b), ReLU outputs of the hidden layer (see Figure 11c), neuron outputs of the output layer (see Figure 11d), and ReLU outputs of the output layer (see Figure 11e), respectively. The legend of each set of graphs is directly corresponding to the net name on the top-level schematic. The amplitude of each graph is annotated in the figure and recorded in a table in the after-mentioned subsection. As seen from the figure, each measuring net has sinusoid outputs, which implies that our MLP circuit can work. In the following subsection, we introduce the performance of our circuit. When comparing Figure 11c to Figure 11b, and Figure 11e to Figure 11d, it was found that the offset of the outputs is eliminated and this is because the capacitor following after the improved source follower filters out the offset.

### 5.2. Summary and Discussion of MLP Circuit

The specifications and weights of our 2-3-4 MLP circuit are summarized in Table 1. Our entire circuit is designed in a 0.6 µm CMOS technology process with supply voltages of ±2.5 V. The working frequency of our circuit ranges from 0 to 1 MHz, which can satisfy the most of applications. In the implementation of the MLP circuit, the interaction of neurons is complicated, such as the impedance issue in cascading and the unit-gain frequency of OPAMP, both can influence the working frequency of the entire circuit. In terms of power dissipation, the measured result shows a power dissipation of 200 mW, as we adopt not an advanced process considering the manufacturing cost. Based on our layout implementation for a perceptron in a 0.6 µm PHENITEC technology, the expected area of the MLP circuit is approximately 1.69 ${\mathrm{mm}}^{2}$.

With respect to the weight value, we list them in the form of two columns by referring to the top-level schematic corresponding to the weight value of neurons in the hidden layer, and neurons in the output layer, respectively. The weight is set to the value consistent with the software by configuring the digital codes of each perceptron. For example, the sixth neuron in our MLP circuit has weights of $w_{36}$ = 3, $w_{46}$ = 0.6, and $w_{56}$ = 0.4, the subscript 3, 4, and 5, represent the starting points, the third, fourth, fifth neuron, respectively. The subscript 6 represents the target point, the sixth neuron. Similarly, the other weights follow the same rule. In this experiment, all biases for neurons are set to 0 for the simplicity of demonstration. However, if bias is enabled in each neuron, we can insert an OPAMP-based adder at the final output of the MLP circuit, adding the amount generated by the weights. Since the bias in each neuron is a constant in the neural network model, the total amount generated by the network is also a constant.

Table 2 shows the amplitude of the measuring nets, their corresponding mathematical calculation, and error ratio. We omit some measuring nets and only list the values at the outputs of the ReLU circuits, as the omitted measuring nets are not final outputs of neurons and also have the offset. In the following, we give an example of the mathematical calculation for a measuring net. For the net N3ReLU_out at the output of the ReLU circuit after the third neuron, which has weights of $w_{13}$ = 2, $w_{23}$ = 1, thus, the output is: $y_{3}$ = $x_{1}$·$w_{13}$ + $x_{2}$·$w_{23}$. Since the amplitudes of $x_{1}$ and $x_{2}$ are both 1 mV in our implementation, the amplitude of $y_{3}$ is 3 mV. Ideally, the amplitude of $y_{3}$ is the final result for the net N3ReLU_out if the circuit can realize the ReLU function perfectly. In the same way, we can calculate that the output of the net N4ReLU_out is 15 mV and that of the net N5ReLU_out is 5 mV. After all of the outputs in the hidden layer is obtained, they are used as inputs of the next layer. The output of neuron in the output layer is equal to that its inputs are multiplied by corresponding weights and are then summed up, the output for the sixth neuron is as: $y_{6}$ = $x_{1}$·$w_{36}$ + $x_{2}$·$w_{46}$ + $x_{3}$·$w_{56}$. Here $x_{1}$, $x_{2}$, and $x_{3}$ denote the inputs of the sixth neuron, the amplitude of $y_{6}$ is 20 mV, the final result for the net VOUT1 is also 20 mV under assumption of the ideal circuit. Similarly, the outputs for the nets VOUT1, VOUT2, and VOUT3, are 13.5 mV, 34 mV, and 10.83 mV, respectively. Note that all the outputs of neurons are positive, as the inverting amplifier inside perceptron module inverts the summed signal. The amplitude for each measuring net is listed in Table 2, and the error ratio is given accordingly. The error ratio is calculated by the difference between the mathematical calculation and simulation result divided by the mathematical calculation.

Furthermore, in order to demonstrate the effectiveness of our analog-based MLP, we configure MLPs with different structures and measure the neuron outputs, power dissertation, and working frequency by using the way as in a 2-3-4 MLP. Finally, analog-based MLPs and comparison to FPGA-based MLPs with the same structures are shown in Table 3. In analog-based MLPs of this work, the structure selections use the geometric pyramid rule and rounding rule as clarified in Section 2, thus the circuit topologies are significantly reduced. Note that the eight cases shown in the table are only for demonstration, as we cannot enumerate all possible structures herein even if we employ the geometric pyramid rule and set a constraint for a die size of 3.6 × 3.6 ${\mathrm{mm}}^{2}$. The best structure can be chosen in a machine learning algorithm by software in our future works. The hidden layer number of cases 1, 2, 7, and 8 is a rounding number as the results by applying the geometric pyramid rule to these cases are not integers.

We also use a specification, RMES (root-mean-square error) of error ratio, to evaluate an analog-based MLP with given structure, which can be obtained by applying RMSE calculation to the error ratio of all neuron outputs (as referred to Table 2), our error ratio is a parameter that characterizes the intrinsic error generated from the analog-based MLP, ideal error ratio for each neuron output is 0. After we obtain a set of error ratio for MLP hardware, the connection weights of the MLP in the software are calibrated to correct values according to the error ratio from hardware. Thus, our learning model from software can achieve a higher precision as compared to the one without considering the hardware error. According to our extensive experience of tuning the connection weights in software, when the RMES of error ratio for a selected MLP hardware is beyond 20%, the selected structure needs to be abandoned as the calibration for the model mismatch in hardware becomes very complicated. To the best of our knowledge, this is the first work that proposes to take the hardware error into account when finding the connection weights.

The specification shows that there is not much difference between the selected different structures. It implies that the system errors resulting from the analog-based MLPs are nearly constant and within the tolerance. For the error ratio, on the one hand, the ideal ReLU function and approximated function generated by the circuit have a difference that is inevitable. On the other hand, the interaction between the analog circuits is complicated, especially in a highly-interconnected neural network, designers need trade-off in the consideration of many aspects, such as noise, offset, impedance, the input, and output range, and the weight value accuracy. The error ratio can be reduced from solving the above issues.

As for power dissipation, analog-based MLPs show that power dissipation varies with the adopted structure, depending on the number of DAC-based multipliers and activation circuits utilized in the MLP circuit. Power dissipation is also different even if in structures with the same neuron number, this is because their multiplier number and activation circuit number are different. Power dissipation for our analog-based MLPs can still be further optimized as our circuits achieve low design complexity and good functionality at the expense of some power. In future works, we can reduce power dissipation by using the following low power design techniques: It is suggested that the OPAMP circuits in a MLP can be replaced by chopper amplifiers with low noise and low power [35,36]. Thus, the signal-to-noise ratio can be improved and power dissipation can be reduced. For the summator block inside an analog perceptron, it can be implemented with lower anlog circuitry using OTAs instead of voltage amplifiers, thus reducing silicon area and power dissipation [40]. Furthermore, aiming at low-power applications, the active resistor comprised of MOS transistors operating in sub-threshold region can achieve high resistance and good tuning capability [41,42,43]. It is suitable for large value resistor implementation for the analog-based MLP.

As for the expected area of a layout, we can obtain a reasonable estimate from the silicon implementation of an analog perceptron. The reason for the difference in power dissipation also applies to the case in the area estimate.

As for the working frequency, our analog-based MLPs with different structures show a wide range of up to 1 MHz, in terms of most common applications that deal with analog signals. This is because the impedance issue of interconnecting neurons has been carefully handled by the buffers inside ReLU modules. We can extend the working frequency range if needed by carefully designing the output impedance of buffer, and the input impedance of analog perceptron. Note that the simulated working frequency demonstrated herein is in the context of no noise present, the noise can be filtered out by a low-pass filter embedded with the MLP, or clean signals are already acquired by filters before the target signals are input into analog-based MLP. Besides, analog-based MLPs have the specification of 4 bits for connection weights. Each connection weight is configured by digital codes of DACs. Lastly, our analog-based MLPs have the merit of low-cost hardware, which facilitates the training for a sensor network where machine learning is applied.

Cases 7 and 8 in analog-based MLPs are used to compare to FPGA-based MLPs in [17,39], respectively. In terms of power dissipation, case 7 consumes less power compared with the best case in [17]. However, case 8 consumes less power considerably compared with the case in [39], this is because the case in [39] utilizes a large number of flip-flops in this structure. As for the comparison to expected area, it is hard to make a rigorous comparison as the FPGA-based MLPs are not implemented in custom chips. However, they provide information that core FPGA chips used for structures in cases [17,39] are 10 × 10 ${\mathrm{mm}}^{2}$ and 17 × 17 ${\mathrm{mm}}^{2}$, respectively. It is significantly larger than ours as even the die size of analog-based MLPs is no more than 3.6 × 3.6 ${\mathrm{mm}}^{2}$. In terms of working frequency and configurable weight bits, FPGA-based MLPs show more advantages than analog-based MLPs. FPGA-based MLPs have very high working frequency, however, a 1 MHz working frequency range is sufficient to most of analog signal applications. The weight value accuracy of FPGA-based MLPs is very high as they employ the fixed-point number, whereas the weight value accuracy of ours is limited. Our analog-based MLPs can be improved by increasing the number of the weight bits. Last, the hardware cost of MLPs based on Xilinx FPGA platform for training or deployment in sensor nodes are very high compared with analog-based MLPs.

Last, it is worth mentioning that when dealing with analog design, process variation becomes one of the critical factors in design. The passive components such as R and C will vary each from 10% to 20%. The lumped variation can be huge, thus harming the circuit function. However, for digital-based design, this issue is avoided. To overcome the stated issue, in future works for our silicon implementation of analog-based MLP, the passive resistors and capacitors can be classified into two types. One has relatively small components whose nominal values are below tens of KΩ or pF. The other has relatively large components whose nominal values are beyond tens of KΩ or pF.

As for the relatively small components, we need to apply some layout techniques to reduce the process variation that degrades the absolute accuracy and matching accuracy. Good practice in compensating the accuracy can be gained from the following rules: Integrated resistors and capacitors are decomposed into many unit cells whose dimension, orientation, and material are identical. The width is 5 times larger than the minimum feature size for high-accuracy resistors, and the perimeter-area ratio is the same for matched capacitors. These unit cells are arranged closely in an array with the same pitch and placed far away from power devices. More precise absolute accuracy and matching accuracy can be achieved by using interdigitated or common centroid structures. For keeping the boundary of all the sides of the capacitor or resistors uniform, dummy strips need to be placed around the components. As for the relatively large components, it is a good choice to use the MOS capacitor and MOS resistor to replace passive capacitor and resistor, respectively. Alternatively, we can use discrete passive components to implement capacitors and resistors with large values in circuitry, since the discrete components generally have a higher absolute accuracy than those implemented in integrated circuits. A more effective method can be referred as in [44], where adjustable resistors and capacitors are implemented in an integrated circuit, based on an external resistor (first reference) and an accurate clock (second reference), one can tune the resistance value and capacitance value to the target percentage of accuracy, respectively.

## 6. Conclusions

This paper presented the measured silicon results for the analog perceptron circuits fabricated in a 0.6 µm/±2.5 V CMOS process, which were comprised of DAC-based multipliers and phase shifters. Based on the measurement-verified perceptron circuits, we implemented a 2-3-4 MLP circuit with ReLU activation, and emulationally demonstrated the feasibility and effectiveness to flexibly configure a MLP-based neural network by utilizing our analog perceptron. The simulation results showed that our MLP circuit had a power dissipation of 200 mW, a range of working frequency from 0 to 1 MHz, and an error ratio within 12.7%. In the implementation of our MLP circuit, an improved source follower was proposed to greatly approximate the ReLU activation function. One summator was adopted to improve the reliability of the whole circuit, so that the DC biasing of the ReLU circuit was adjustable. In addition, the impedance issue in the cascading of neurons in MLP was presented, which is critical to the feasibility of the whole circuit. Moreover, seven more analog-based MLPs designed with the same approach were used to analyze the simulation results with respect to various specifications, in which two cases were used to compare to their digital counterparts with the same structures. The results measured from the simulation are summarized and analyzed in this work. As prototyping circuits for the proof-of-concept of on-chip learning, our circuits showed issues with respect to noise, resistance accuracy, and prediction accuracy. Optimizing these specifications is one of our future works.

## Figures and Tables

**Figure 1 sensors-20-04222-f001:**
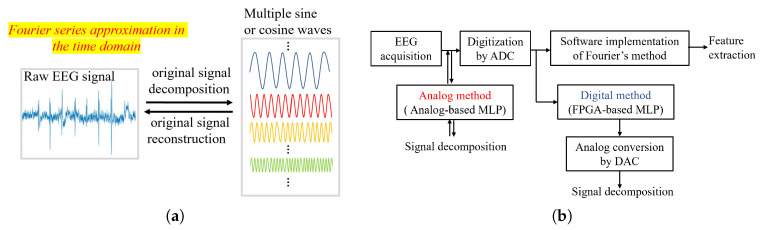
Comparison between the analog method and digital method aimed at electroencephalogram (EEG) signal processing with machine learning mechanism. (**a**) EEG signal decomposition/reconstruction by using fourier series approximation. (**b**) The analog-based multi-layer perceptron (MLP) and field programmable gate array (FPGA)-based MLP in the handling of the EEG siganl.

**Figure 2 sensors-20-04222-f002:**
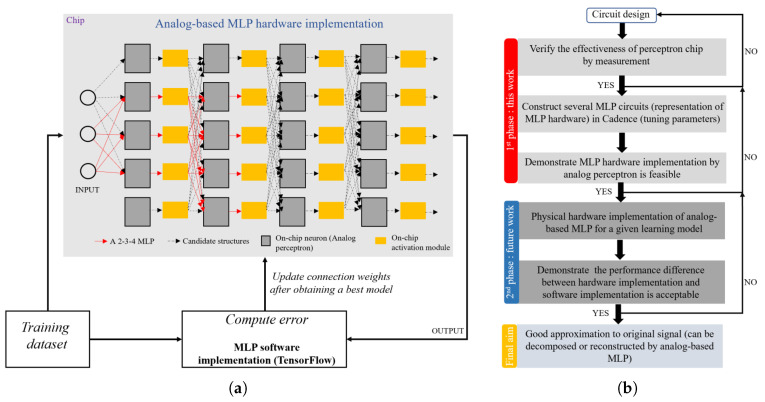
(**a**) Configuration of our proposed analog-based MLP and learning algorithm. (**b**) Contributions in this work.

**Figure 3 sensors-20-04222-f003:**
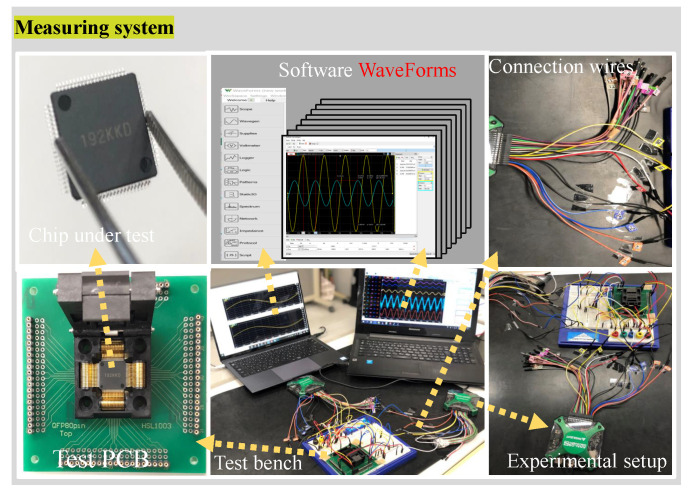
The general architecture of the measuring system for the perceptron chip.

**Figure 4 sensors-20-04222-f004:**
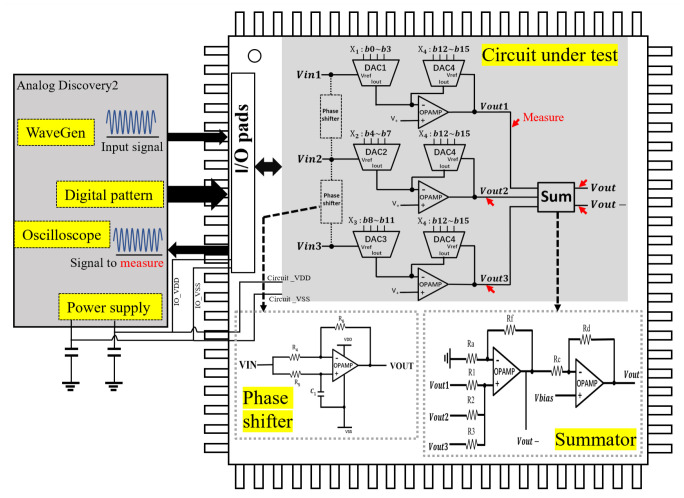
A schematic of a perceptron chip and block diagram of the measurement.

**Figure 5 sensors-20-04222-f005:**
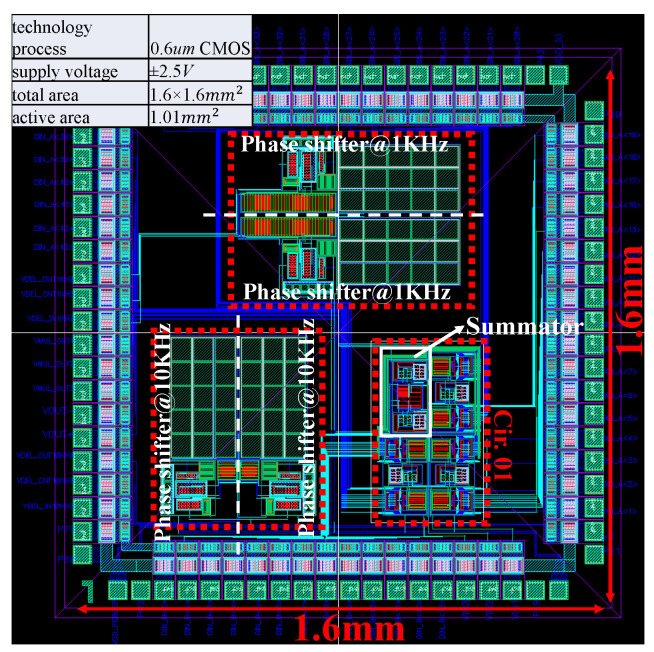
Top-level layout of perceptron chip.

**Figure 6 sensors-20-04222-f006:**
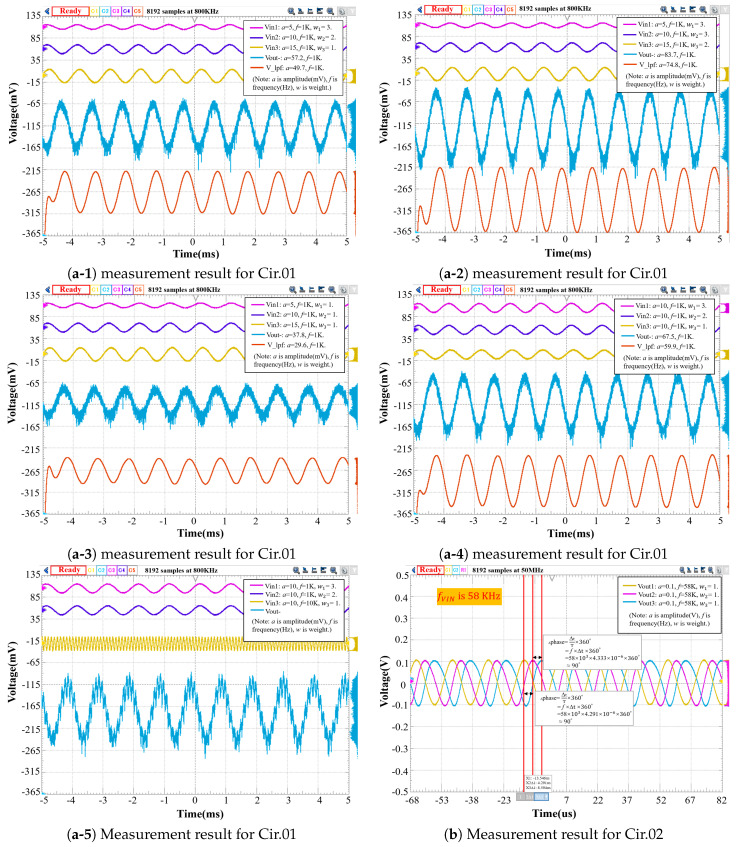
Measurement results for perceptron chip.

**Figure 7 sensors-20-04222-f007:**
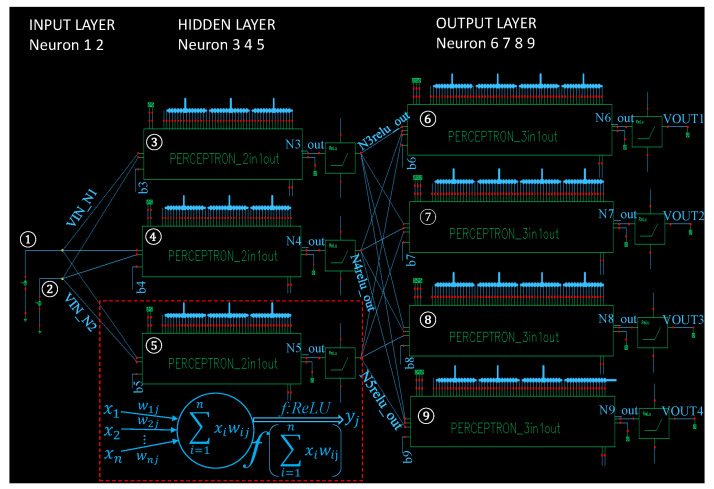
Top-level schematic of our MLP circuit.

**Figure 8 sensors-20-04222-f008:**
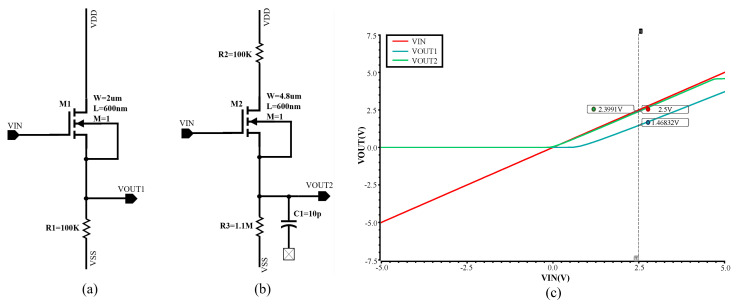
Circuits and simulation for the rectified linear unit (ReLU) activation function. (**a**) A source follower circuit. (**b**) Our improved source follower circuit. (**c**) Simulation results for both circuits.

**Figure 9 sensors-20-04222-f009:**
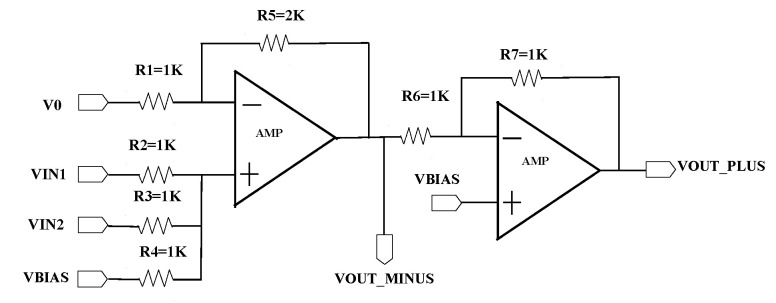
Summator inside our perceptron module.

**Figure 10 sensors-20-04222-f010:**
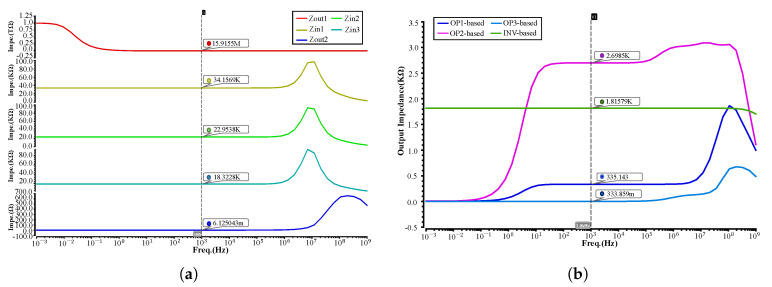
Impedance issue of cascading. (**a**) Comparison between the output impedance and the input impedance. (**b**) The output impedance of the circuit after inserting buffers.

**Figure 11 sensors-20-04222-f011:**
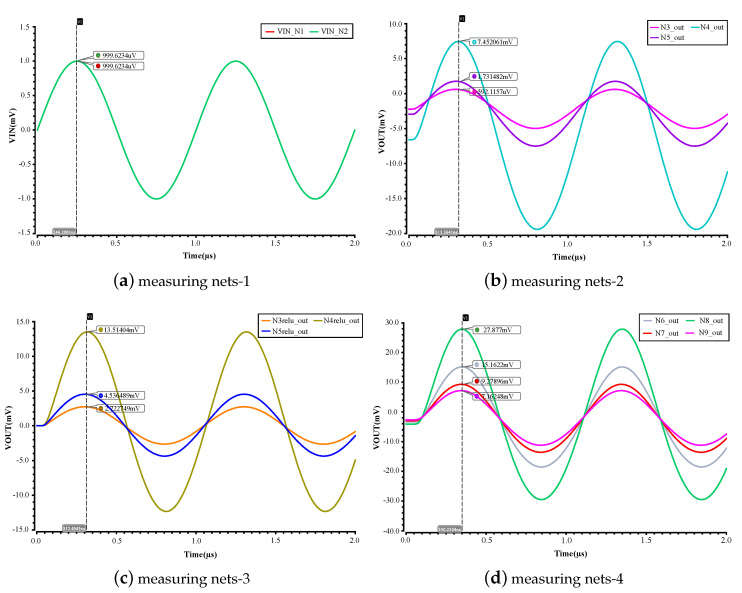
Simulation results for the top-level schematic.

**Table 1 sensors-20-04222-t001:** The specifications and weights of our 2-3-4 MLP circuit.

Technology process	0.6 μm CMOS	
Supply voltage	±2.5 V	
Working frequency	0–1 MHz	
Power dissipation	200 mW	
Expected area	1.69 ${\mathrm{mm}}^{2}$	
Weight value	$w_{13}$ = 2, $w_{23}$ = 1	$w_{36}$ = 3, $w_{46}$ = 0.6, $w_{56}$ = 0.4
	$w_{14}$ = 5, $w_{24}$ = 10	$w_{37}$ = 2, $w_{47}$ = 0.25, $w_{57}$ = 0.75
	$w_{15}$ = 3, $w_{25}$ = 2	$w_{38}$ = 3, $w_{48}$ = 1, $w_{58}$ = 2
	n/a	$w_{39}$ = 0.83, $w_{49}$ = 0.16, $w_{59}$ = 1.16

**Table 2 sensors-20-04222-t002:** Evaluation to our MLP circuit.

Measuring Net	Mathematical	Simulation	Error
	Calculation (mV)	Result (mV)	Ratio (%)
N3ReLU_out	3	2.72	9.3
N4ReLU_out	15	13.51	9.9
N5ReLU_out	5	4.54	9.2
VOUT1	20	17.81	11.0
VOUT2	13.5	11.79	12.7
VOUT3	34	30.79	9.4
VOUT4	10.83	9.45	12.7

**Table 3 sensors-20-04222-t003:** Analog-based MLPs and comparision to FPGA-based MLPs with same structures.

	Case	Structure	RMSE(%)	Power	Expected	Working	Configurable	Hardware
MLP			of	Dissipation	Area	Frequency	Weight Bits	Cost
			Error Ratio	(mW)	(${\mathrm{mm}}^{2}$)	(MHz)	(bit)	
This work	1	2-3-4	10.70	200	1.69	1	4	Perceptron chip (low)
	2	4-3-2	10.73	192	1.56			
	3	1-2-4	10.40	131	1.17			
	4	4-2-1	10.33	118	0.96			
	5	1-3-9	10.59	223	2.73			
(Analog-based MLPs)	6	9-3-1	10.41	219	2.17			
	7	7-6-5	10.58	234	3.05			
	8	12-3-1	10.37	228	2.68			
FPGA-based MLPs	[17]	7-5-6	-	241	-	100	16	Artix-7 (high)
	[39]	12-3-1	-	1776	-	100	24	Zynq-7000 (high)

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
