# Peer review of "Implementation of Analog Perceptron as an Essential Element of Configurable Neural Networks"

_sensors, 2020, doi:10.3390/s20154222_

Round 1
Reviewer 1 Report
- Title of the paper does not precisely describe the presented research results. My recommendation for the title would be: "Implementation Study of Perceptron as an Essential Element of Configurable Neural Networks"
- Introduction should explain the main idea, which is a silicon implementation of perceptron and how to build a configurable neural network of it.
- Paper structure confuses the reader (implementation and measurement results before the concept). Namely, one should start with description of the concept (in this case, a multi-layer perceptron based CNN). Then, the implementation and measurement results can be presented. Please exchange the order of Section 2 and Section 3.
- Finally, there is no clear justification/explanation of the configuration selection made: why have the authors selected a 2-3-4 MLP configuration? The choice of other neural network parameters (weighting coefficients and activation functions) is also not discussed.
Reviewer 2 Report
This manuscript needs more details on recent implemenations of perceptrons not only in the analog domain but also in the digital one, as shown in the recent paper : FPGA-based implementation of a multilayer perceptron suitable for chaotic time series prediction, where tha authors introduce a multilayer perceptron that is implemented using a field-programmable gate array (FPGA) to predict experimental chaotic time series. One can see that the topology of an atificial neural network ca ne implemented by appling the geometric pyramid rule, but it produces a very large topology and some neurons can be deleted as the authors in this paper show. You need to describe the generation of the proposed topology and detail how did you get to this topology and the possibilities of implementation.You mention that: To demonstrate the feasibility and effectiveness of our MLP circuit, We give an example of a 2-3-4 MLP circuit with rectified linear unit (ReLU) activation, which consists of 2 input neurons, 3 hidden neurons, and 4 output neurons. Its experimental case shows that the simulated performance achieves a power dissipation of 200mW, a wide range of working frequency from 0 to 1MHz, and a moderate performance in terms of the error ratio... Why you use just three layers and why an increasing number of neurons from 2 to 4?. Also, you get up to 1MHz of working frequency, but as mentioned in th suggested paper above, using digital hardware as the FPGA one can reach a more higher frequency of operation and the neurons can be reprogrammed/configured. You my discuss the usefulness of implementing an anlog integrated circuit instead of using digital electronics.
You use amplifiers as shown in Figure 2. A schematic of a perceptron chip and block diagram of the measurement, but the adder, for example, can be implemented with lower anlog circuitry using operational transconductance amplifiers (OTAs) instead of voltage amplifiers thus reducing silicon area and power consumption. You can review several pepers on OTAs as for exmaple in: Pseudo-Three-Stage Miller Op-Amp With Enhanced Small-Signal and Large-Signal Performance, where you can see the necessity of comparing performances by using a figure of merit. This may be a future work but using OTAs you can eliminate the voltage sum-node and then reduce silicon area.
You just mention that many digital- or analog-implemented circuits for perceptron have been proposed in the literatures, which show the good results in the simulation phase [11–16]. But more details are required with respect to advantages in the analog domain, and also you may discuss issues on applications where the analog circuitry may be of great importance. One hot topic of application of artificial neural netowrks is cryptography, as shown in the recent reference: Chaotic Image Encryption Using Hopfield and Hindmarsh–Rose Neurons Implemented on FPGA, again, you can see that those kind of neurons are implemented on digital hardware, so that you need to motivate the reader why you are working on the analog domain and what will be the main benefit of analog electronics compared to the digital one.
You also need to discuss the possibilities of changing the characteristics of the neurons, the weights, activation function, training, online or offline and the possibilities when using analog electronics. In Section 3 you mention something about this: By utilizing the analog perceptron chips and activation circuits with some electronic components, after fine-tuning, we can construct MLP-based neural networks with any number of layers and neurons per layer. Compared with the on-chip learning circuit whose network is integrated into a single chip, the configurability and flexibility of our method are improved, which is helpful for our circuit devoted to analog front-end of the biomedical sensor. The number of layers and neurons per layer as well as component values can be determined by circuit simulation... but it is not clear how did you get to the number of neurons in each layer, peforming simulations is good but you may do infinite loops. Can you provide more references or details on how to choose the number of layers and neurons in each layer?
In the same Section 3 you mention: In this section, a multi-layer perceptron circuit with a non-linear activation function is introduced... why did you choose this activation function, why not a linear function? or the hypherbolic one? You can see the reference suggested above where the activation function is of hypherbolic type, and which can be esaily implemented using OTAs, while in the output neurons sometimes it is enough by using a linear activation function, and it reduces design efforts.
In Figure 6. Circuits and simulation for the ReLU activation function, you use resistors, and again, they need large silicon area but you may review its silicon implementation with novel techniques, search for example the silicon implementation of large value resistors, and again, sometimes the OTA can be used to implement resistors, a good tutorial on OTAs is given in reference: Active Filter Design Using Operational Transconductance Amplifiers: A Tutorial.
Reviewer 3 Report
Review of “Configurable Neural Network Based on An Implementation-on-silicon Perceptron”
Overview
The paper presents a fabricated chips that implement a multi-layer perceptron with ReLU neurons, acting with analog inputs and outputs and capable of handling input frequencies of up to ~1MHz. They demonstrate performance with a 2-3-4 Perceptron, and report performance of within 10% of the values that would be produced with an ideal mathematical model.
Positives
I like that they are building analog neural networks like this, and I definitely think more work needs to be done in this sort of direction. I also like the amount of detail they’re reported here in terms of the analysis process and the hardware used. The basic task is clear and well-described.
I also like the explicit look at the analog error (final column of Table 2).
Negatives
I’m still confused a little bit about what the final performance of the system is. Figure 9 shows simulation results for the 2-3-4 perceptron, but why is there no corresponding figure for the results from the physical hardware? Instead, there’s just the summary information in Table 2.
More importantly, I think I’m also confused as to what sort of comparison they’re thinking of when evaluating the chip. They say “it’s still far smaller than most of FPGAs(field-programmable gate arrays)- or GPUs(graphics processing units)-based implementations.” How much smaller? And what capabilities are they comparing that to? My best guess is that they are comparing to a digital 2-3-4 ReLU perceptron capable of operating at ~1MHz with 16-bit configurable weights. Is that the right point of comparison? How big would a custom ASIC for that operation need to be? It would also be great to have estimates for what the energy consumption on a small GPU or FPGA would be. (These would not have to be detailed implementations, but just something to give a ballpark number and justification for statements like “far smaller” and “moderate performance” would be very useful.)
Also, if I am right that they’re thinking of comparing to a digital implementation running at ~1MHz, then one big important thing about a digital implementation is that there’s no temporal filter, and you don’t have to worry about any temporal effects at all. Figure 8 does a great job of showing that things should be fine, but if so then I think it’d be very important to show an example of it working with inputs that aren’t just phase-locked sine waves of the same frequencies. Does it work with band-limited white noise inputs? That seems to me that it’d be closer to the sorts of inputs that such a system might need to handle when deployed.
Finally, and perhaps more controversially, while I found the methods they used to match the ReLU behaviour fascinating, I would also like the authors to discuss more about why they want to match ReLU exactly. The only reason ReLU is used in digital machine learning is that ReLU is very easy to compute on digital hardware. But there’s nothing special about it: the simpler source follower (Figure 6a) is a perfectly fine component to use for a multi-layer perceptron. They could just use a that slightly shifted function for the software simulation, and everything should work fine. The only thing a neural network needs is *some* non-linearity, and one of the exciting things for me about analog hardware is that there are all sorts of very-easy-to-implement non-linearities that are available, and there’s no particular reason to use exactly the same non-linearities that digital machine learning uses. So I'd like to know why the authors took this approach. Is the goal to eventually take standard published networks and implement them on this sort of hardware? Or is this more of a "just to see if we can" sort of thing?
Small points for clarification:
How many bits for the weight specification?
Was the 2-3-4 perceptron implemented as one chip or multiple chips? How many chips?
Line 44: What does this mean: “By utilizing all of the available perceptron chips”
Was temperature variability an issue?
Couple you do the back-prop training on a simulated version that was calibrated to the mismatch in the model? In other words, could you take the error shown in table 2 into account when finding connection weights?
Line 335: “200mV” should be “200mW”
Round 2
Reviewer 1 Report
1. Please enlarge Figure 5 and Figure 7. In the present form, they are not readable.
2. Please provide a short introduction for Section 2, 3, and 5 (as you did in Section 4).
3. Please improve the language.
Reviewer 2 Report
the revised manuscript can be accepted in its present form
Author Response
We appreciate that you put lots of efforts in this work so that we can improve our manuscript and our research.
Reviewer 3 Report
Thank you to the authors for their revisions. The new version is much clearer. The added Section 2 helps a lot to situate the work and indicate where my assumptions were wrong! I also very much like Table 3, and that addresses my concerns about what the points of comparison are (including the number of bits in the weights)
As for how the system responds to white noise input, I think the author's answers in the response (that this is for future work) make sense. However, I still disagree with the authors that "ReLU is employed because it shows more advantages than other functions in terms of gradient vanishing problem, convergence speed, computation efficiency." The shifted ReLU that they show in Figure 8 will have exactly the same vanishing gradient and convergence speed as the standard ReLU, and it will be more computationally efficient as doesn't require the extra resistor and capacitor (and uses a smaller resistor). But, this is a pretty controversial point that I'm making, and I completely understand sticking with the standard ReLU that people are familiar with. So I don't think this is a point that needs to be addressed in revision.
Overall, the authors have addressed all the issues that I think need to be addressed. Thank you!
Author Response
Honorable SENSORS’ s reviewer,
We appreciate that you put lots of effort in this work so that we can improve our manuscript and our research. We also like your discussion with us.
Here, we would like to clarify that in the manuscript, we didn't mean that the ReLU circuit employed in this work shows more advantages than other functions including the standard ReLU, we intended to mean "the standard ReLU shows more advantages than other functions in terms of gradient vanishing problem, convergence speed, computation efficiency." We set the standard ReLU as our goal to approximate, the realized function by circuit has been very close to the standard ReLU.
Maybe our description for this is not clear in the original revised version. To make it more clearly, we added some sentences in a newly-reviewed version, you can check it.
Best regards,
Chao Geng